# The Prevalence and Characteristics of Diffuse Idiopathic Skeletal Hyperostosis (DISH): A Cross-Sectional Study of 1519 Japanese Individuals

**DOI:** 10.3390/diagnostics12051088

**Published:** 2022-04-27

**Authors:** Hisanori Ikuma, Tomohiko Hirose, Dai Nakamura, Kazutaka Yamashita, Masataka Ueda, Kazuhiro Sasaki, Keisuke Kawasaki

**Affiliations:** 1Department of Orthopaedic Surgery, Kagawa Prefectural Central Hospital, 2-1, Asahi-Machi 1-Chome, Takamatsu 760-8557, Japan; tmhkhrs@yahoo.co.jp (T.H.); d.nakamura.1167@gmail.com (D.N.); p90v6srf@s.okayama-u.ac.jp (K.Y.); muscat4ade@gmail.com (M.U.); kkeisuke@a1.mbn.or.jp (K.K.); 2Department of Emergency and Critical Care Medicine, Kagawa Prefectural Central Hospital, 2-1, Asahi-Machi 1-Chome, Takamatsu 760-8557, Japan; kazugreggu@ybb.ne.jp

**Keywords:** diffuse idiopathic skeletal hyperostosis, prevalence, Japanese individuals, hyperostosis, costovertebral joint

## Abstract

Background: Diffuse idiopathic skeletal hyperostosis (DISH) is a pathology characterized by enthesis ossification, but there have been few reports on epidemiological surveys. This report presents a cross-sectional survey of DISH from thoracic to sacral spine in patients at the tertiary emergency medical center. Methods: The patients were divided into DISH (+) group and DISH (−) group for a retrospective comparative study. The primary outcome measures were the frequency of DISH and the patient demographic data, the secondary outcome measures were the previous medical history (diabetes mellitus, cardiovascular disease), the extent of aortic calcification, the frequency of hyperostosis around the costovertebral joint and the mortality rate within 3 months of the initial examination. Results: This survey examined a total of 1519 patients. There were 265 cases (17.4%) in the DISH (+) group and 1254 cases in DISH (−) group. The prevalence of DISH was concentrated at the thoracolumbar junction, particularly at T9. The mean age, ratio of male and hyperostosis around the costovertebral joint were significantly higher in the DISH (+) group (*p* < 0.001), but there was no significant difference in other variables. Conclusions: The pathology of DISH might involve the effects of age-related changes or biomechanical effects.

## 1. Introduction

Diffuse idiopathic skeletal hyperostosis (DISH) is a systemic condition that is characterized by the ossification of enthesis, and in the spinal column in particular, there may be spinal ankylosis or hyperostosis in the antero-lateral side of the vertebral body, occurring in four or more consecutive vertebral bodies [1,2]. Additionally, the prevalence of this disease will increase in elderly patients. In recent years, as populations grow older throughout the world, there has been a tendency for increased frequency of cases associated with DISH in the spinal column [3,4]. This disease will cause ankylosing changes in the spinal column, resulting in decreased spinal flexibility and the global balance [5]. Ultimately, this can make the patient susceptible to falls, leading to spinal fractures [6,7,8]. Furthermore, if there is a severe ankylosing of the thoracic spine, hyperostosis around the costovertebral joint may occur, increasing the possibility of decreased respiratory function. In the case of a spinal cord injury, the fusion of the costovertebral joint can further exacerbate the respiratory state, shortening the vital prognosis [9]. Additionally, when looking at bone metabolism, the spinal ankylosis can cause a stress shielding in the vertebral body in the area of the ankylosis, and it is known that there will be a reduction in bone density within the vertebral body in inverse proportion to the hyperostosis occurring outside of the vertebral body over the course of time [10,11]. When attempting to evaluate bone mineral density using dual-energy X-ray absorptiometry (DEXA), the hyperostosis occurring outside of the vertebral body can result in an apparent increase in bone mineral density, making it difficult to evaluate the osteoporosis within the vertebral body. Therefore, this may prevent the adequate treatment for osteoporosis in the case with DISH, making the patient susceptible to have a spinal fracture. As described above, many of the issues that may be observed in the spinal column associated with DISH have been noted to date, but there are few reports of epidemiological surveys of this pathology. In this study, we report on a cross-sectional survey on DISH under thoracic spine that was conducted on patients that were examined as outpatients at the tertiary emergency medical center of our institute.

## 2. Materials and Methods

### 2.1. Patients

All of the patients that were examined with the torso computed tomography (CT) imaging as outpatients at the tertiary emergency medical center of our institute from January 2020 to December 2020 were included regardless of their diagnosis in this cross-sectional study on DISH. Because the majority of patients examined as emergency outpatients at our institute will undergo routinely CT imaging from the chest to pelvis, the vertebral segments to be investigated were set from the first thoracic vertebra to the first sacral vertebra, and patients for whom there had been the multiplanar reconstruction (sagittal, axial, and coronal plane) in the torso CT imaging at the time of the initial examination were extracted. Due to the small number of cases for which imaging was performed up to the pelvis including the cervical spine, the decision was made to exclude the cervical spine from the target of this study. These patients were divided into two groups based on the existence of DISH (a DISH (+) group and a DISH (−) group), after which a retrospective investigation was conducted on the following parameters. The patient demographic information (age, gender and body mass index (BMI)), the previous medical history (diabetes mellitus, cardiac or vascular disease) and the mortality rate within 3 months of the initial examination were investigated from the medical charts of patients. As the radiographic data, the extent of aortic calcification and the frequency of hyperostosis around the costovertebral joint were collected. The level and distribution of DISH were also investigated in the DISH (+) group for a sub-study.

### 2.2. Imaging Evaluation of DISH

The sagittal, axial and coronal plane in the CT images were evaluated with the criteria by Resnick and Niwayama et al. [1] as reference in order to assess the presence of DISH. Cases showing bone bridging over at least four consecutive vertebral levels on the anterior or lateral side of the vertebral body as well as preservation of intervertebral disc height and showing no inflammation or ankylosing changes in the sacroiliac joint were judged to have DISH. In this study, the existence of DISH was evaluated irrespective of the stage of ossification progress. The images were evaluated by four orthopedic surgeons (two residents and two senior spine surgeons) who were blinded to the actual clinical course of the patients.

### 2.3. Evaluation of the Aortic Calcification

The calcification of the vascular walls was evaluated in the aorta. Patients were scanned in the supine position in the craniocaudal direction, using a 64-slice non-contrast CT scan (General Electric Company, Boston, MA, USA), in which images were obtained with a 10 mm single slice thickness. Calcification was considered to be present if an area of ≥1 mm^2^ displayed a density of ≥130 Hounsfield units. The calcification of aorta was calculated from the distal end of aortic arch to the bifurcation of the aorta into the common iliac arteries. For the evaluation of the vascular wall calcification, a semi-quantitative measurement of the calcification of the aorta was conducted independently by two residents and two senior spine surgeons who were blinded to the patient’s clinical characteristics. The cross-section of the aorta on each slice was divided radially into 12 segments [12]. The aortic calcification index (ACI) was calculated as follows: ACI = (total score for calcification in all slices)/12 × 1/number of slices) × 100 (%) [13,14] (Figure 1). In this study, cases showing an ACI value of 50% or more were judged to show significant calcification of the aortic vascular walls, and the frequency of the occurrence of these cases was compared between the DISH (+) group and DISH (−) group.

### 2.4. Previous History of Cardiac or Vascular Disease

The existence of a previous medical history of the following cardiac or vascular diseases was investigated and the frequency of these medical histories was compared between the DISH (+) group and DISH (−) group.

Previous history of cardiac or vascular disease: cerebral infarction, cerebrovascular hemorrhage, cardiac valvulopathy, angina pectoris, myocardial infarction, atrial fibrillation, chronic heart failure, chronic renal failure, hemodialysis, hypertension, pulmonary thromboembolism, arteriosclerosis obliterans, aortic aneurysm and aortic dissection.

### 2.5. Evaluation of Hyperostosis around the Costovertebral Joint

The CT imaging of axial plane showing the area from the T1 vertebra to the T12 vertebra was used in this evaluation. In reference to the report by Sawagami et al. [9], the ankylosing or bone bridging of the costovertebral joint was recorded when the complete fusion and osseous outgrowth (ossification of the radiate ligament of the head of the rib) was apparent at the level of the costovertebral joint. Cases showing these findings in even one site within the area from the T1 vertebra to T12 vertebra were judged to show the costovertebral fusion, and the number of these cases was compared between the DISH (+) group and DISH (−) group (Figure 2).

### 2.6. Ethical Considerations

We performed this investigation in accordance with our institutional guidelines that comply with international laws and policies (Institutional Review Board of Kagawa Prefectural Central Hospital, IRB approved no. 1081). Informed consent was obtained from all individual participants included in this study.

### 2.7. Statistical Analysis

In this study, we conducted a statistical study using EZR [15] (Saitama Medical Center, Jichi Medical University, Saitama, Japan), which is a graphical user interface for R (The R Foundation for Statical computing, Vienna, Austria). More precisely, it is a modified version of R commander designed to add statistical function frequently used in biostatistics. All measures were compared using independent samples t-tests and chi-square analyses as appropriate, with significance set at *p* < 0.05. For matching of samples in the DISH (+) group and DISH (−) group, the propensity score matching is used with ages and gender treated as confounding factors. These matched samples performed the statistical significance test using the Mantel–Haenszel test.

## 3. Results

### 3.1. Research Subjects and Demographics

A total of 8512 patients were examined as outpatients at the tertiary emergency medical center of our institute during the period from January 2020 to December 2020. Of this total, the patients that underwent CT imaging from the chest to pelvis were extracted and data for the same patients obtained on different examination days were deleted, leaving a total of 1519 patients to be investigated in this study. Of these patients, there were 265 cases (17.4%) in the DISH (+) group and 1254 cases (82.5%) in the DISH (−) group. DISH occurred in all of the segments, from the first thoracic vertebra to the first sacral vertebra. The frequency of occurrence of DISH was higher in the thoracic spine than in the lumbar spine, and the thoracolumbar junction was the most common area in which DISH was found. The prevalence of DISH was particularly high (11.1%) in T9 vertebra. Bone bridging was observed in 8.2 ± 3.2 vertebrae on average, and there were nine cases (3.3%) that showed the double DISH which DISH was found in two areas in one patient (Figure 3).

The mean age of the patients was 78.8 ± 10.1 years in the DISH (+) group and 67.8 ± 19.5 years in the DISH (−) group, and the patients in the DISH (+) group were significantly older compared to the DISH (−) group (*p* < 0.001). Regarding the gender ratio, there were 185 males (69.8%) and 80 females (30.2%) in the DISH (+), and 646 males (51.6%) and 608 females (48.5%) in the DISH (−) group, with a significantly higher ratio of males in the DISH (+) group (*p* < 0.001). The mean BMI was 22.1 ± 4.3 kg/m^2^ in the DISH (+) group and 23.4 ± 3.7 kg/m^2^ in the DISH (−) group, and there was no significant difference between the groups (Table 1). From these results, there was a statistically significant difference in age and gender among the patients’ demographic data in these two groups. Therefore, the propensity score matching was used to perform the sample matching with age and gender as confounding factors. Ultimately, the DISH (+) group contained 188 cases (mean age: 78.8 ± 9.5 years, 65.4% male, mean BMI was 22.2 ± 4.4 kg/m^2^), while the DISH (−) group contained 188 cases (mean age: 79.3 ± 9.5 years, 62.2% male, mean BMI was 20.5 ± 4.0 kg/m^2^).

The previous medical history (diabetes mellitus, cardiac or vascular disease), extent of aortic calcification, the frequency of hyperostosis around the costovertebral joint and mortality rate within 3 months of the initial examination were compared between the DISH (+) group and DISH (−) group. The Standardized Mean Difference (SMD) between these two groups was 0.711 for the age and 0.387 for the gender prior to matching, respectively, while the SMD after matching had improved to 0.060 for the age and 0.066 for the gender. It was judged that there had been adequate matching of the groups from this result (Table 2).

### 3.2. Previous Medical History

Patients with a history of diabetes mellitus comprised 22.9% of the DISH (+) group and 18.1% of the DISH (−) group, while patients with past cardiovascular disease comprised 50.5% of the DISH (+) group and 45.7% of the DISH (−) group, and there was no significant difference in these values between these groups (Table 3). The details of the previous medical history were shown in Table 4.

### 3.3. Extent of Aortic Calcification

The mean ACI was 46.4 ± 30.0% in the DISH (+) group and 47.2 ± 30.6% in the DISH (−) group. The frequency of cases showing an ACI value of at least 50% and judged to have significant calcification was 49.5% in the DISH (+) group and 51.1% in the DISH (−) group, with no significant difference observed between the groups (Table 3).

### 3.4. Frequency of Hyperostosis around the Costovertebral Joint

The prevalence of hyperostosis around the costovertebral joint was 45.7% in the DISH (+) group and 6.3% in the DISH (−) group, and the prevalence was significantly higher in the DISH (+) group, indicating that hyperostosis around the costovertebral joint is a characteristic finding of DISH (Table 3).

### 3.5. Mortality Rate within 3 Months of the Initial Examination

The mortality rate within 3 months after the outpatient emergency examination at this facility was 10.6% in the DISH (+) group and 13.3% in the DISH (−) group, with no significant difference between the groups (Table 3).

## 4. Discussion

### 4.1. Issues with DISH

DISH was first reported by Forestier et al. [16] in the 1950s, after which Resnick et al. expressly advocated for an imaging-based diagnostic method [17]. A spinal column with DISH is characterized by ankylosis spread over a minimum of four vertebral levels with ossification in the anterior or lateral side of the vertebral body, though there may be a lack of findings of sacroiliac joint ankylosis, osteosclerosis or erosion [1]. As a result, there may be a marked reduction in spinal column flexibility, which may impact daily activities, leading to a decreased quality of life. However, the cause of this disease is still not well understood, and the only reports to discuss the issue have noted that the disease is frequently observed in elderly individuals, patients with a history of type 2 diabetes mellitus or obese patients [18,19,20,21]. On the other hand, decreased spinal column flexibility due to ankylosing changes can increase the possibility of falls or fractures, and with spinal fractures in the ankylosis segment, stress may be easily concentrated at the fracture site [22]. This causes the major dislocation of the fracture site, leading to spinal cord injury at times. These issues make it difficult to serve a reliable treatment for spinal fracture accompanied by DISH [10,23,24].

### 4.2. Course of the Epidemiological Survey on DISH

To date, there have been some reports on epidemiological surveys on spinal columns with DISH, but there have been few reports on the epidemiology of Japanese patients as an Asian population. Further, there are many issues associated with DISH as discussed above, so the decision was made to conduct this survey on DISH in hopes of understanding the characteristics of this disease and aiding future treatments. The decision was made to look at patients examined as emergency outpatients because this is a patient group that shows no bias to the orthopedic surgery, which would help to minimize any disease bias, and because the high frequency of CT imaging of the torso during these examinations facilitated enrollment into this study. In this study, a cross-sectional survey was conducted on a total of 1519 patients who underwent torso CT imaging during examinations as emergency outpatients at our facility during a one-year period, and the resultant DISH prevalence was 17.4% (265 cases). Previous reports [25,26,27] have indicated a DISH prevalence of 2.9~42.0%, so the result in this study was consistent with these past reports. Hasegawa et al. reported the prevalence of 27.2% for DISH on CT imaging in Japanese individuals at any department other than orthopedic surgery [28]. However, this study found the prevalence of 17.4% on CT imaging. We consider that the difference in these results would be affected by the disease bias of emergency outpatients and the lack of testing for intra- and interobserver error (one time evaluation, and two residents and two senior spine surgeons for observation) in this study. Additionally, the ratio of elderly individuals and males in the DISH group was significantly higher than in the group without DISH, and this result matched the results reported in a number of papers [18,28,29,30,31,32,33]. However, there was no significant difference in the BMI between the two groups. The reports to date had indicated that there was a significant relationship between DISH and obesity [5,25,30,34,35], but the result from this study could have been impacted by the fact that this study looked at Asians, who show a comparatively low frequency of severe obesity, or the fact that both groups contained a comparatively large number of elderly patients.

### 4.3. Range and Levels of DISH

DISH was observed over a broad range, extending from the first thoracic vertebra to the first sacral vertebra, but the segment showing the highest frequency of findings was the thoracolumbar junction, with a concentration at the T9 vertebra. Additionally, DISH was more frequently observed in the thoracic spine than in the lumbar spine. The past reports as well noted that the highest prevalence was in the thoracic spine, so the results of this study were consistent with the literature. Although the thoracic spine features more joints than the lumbar spine, including facet joints and costovertebral joints, it is characterized by extremely limited mobility because of the structure of the rib cage. Therefore, it may be more susceptible to ankylosis due to age-related changes than the lumbar spine which has a significant mobility. The bone bridging was observed over an extremely broad range, with an average value of 8.2 vertebrae.

### 4.4. Patient Matching

The majority of the reports involved comparative studies with direct classification of patients into a DISH group or a group without DISH, or were case–control study, and there have been relatively few reports that included matching of the patient age or gender. In this study as well, the target population was simply divided into a DISH (+) group and a DISH (−) group, and the results were consistent with past reports in that the DISH (+) group was significantly older and contained a significant large number of males. Of the parameters that were investigated, it had been thought that age or gender would have a significant impact on the parameters of the previous medical history (diabetes mellitus, past cardiovascular disease), extent of aortic calcification, ratio of costovertebral synostosis and mortality rate within 3 months of the initial examination. For this reason, the propensity score matching was used to prepare two new groups in order to exclude the effects of age and gender in this study, and these parameters were compared again.

### 4.5. Relationship between DISH and Diabetes Mellitus

A report on the ossification of posterior longitudinal ligament (OPLL) noted that there was a significantly large number of cases with a previous history of diabetes mellitus in the case with OPLL, while a detailed relationship between DISH and diabetes mellitus has not been clarified. Okada et al. analyzed 177 cases of spinal injury associated with DISH and reported that the prevalence of diabetes mellitus was 28.8%, but the causal relationship with DISH is unknown [36]. Some reports claimed to have discovered a significant relationship between DISH and diabetes [5,35,37] while others claimed to have discovered no relationship with this pathology [30,34,38] and no definite conclusions have been found yet. In this study, the frequency of patients with a previous history of diabetes mellitus was investigated, but there was no significant difference between the DISH (+) group and the DISH (−) group.

### 4.6. Prevalence of Cardiac or Vascular Events

There have been reports on the relationship between DISH and vascular occlusion in recent years. Though some reports have suggested that there is no relationship between the prevalence of ischemic heart disease and DISH [20,30], other reports with matching age or BMI have mentioned that the prevalence of aortic sclerosis or calcification was significantly higher in the DISH group than in the group without DISH [31,34]. Similarly, it was reported that the prevalence of coronary artery calcification was significantly higher in the DISH group than in the group without DISH [39]. However, in this study, there was no significant difference between the groups in the frequency of patients with an ACI of 50% or more or in the prevalence of past cardiovascular disease, and it was not possible to discover a relationship between the previous cardiovascular events and DISH. In this way, it was not possible in this study to detect the possibility that DISH has a relationship with the glucose metabolism that has been well mentioned in reports on OPLL. It is very likely that the pathology of DISH is impacted by age-related changes or biomechanical effects, and there may be a possibility that DISH has a different pathology compared to OPLL which is based on genetic or molecular biological factors.

### 4.7. Mortality Rate within 3 Months of the Initial Examination

In this study, we set the period of the mortality rate to 3 months after an admission, because the clinical courses of patients who hospitalized with a severe general condition at the tertiary emergency medical center are often aggravated within 2 months; moreover, the mean age of the patients that were included in this study were over 65 years of age. Sawagami et al. reported that cases with DISH often show the hyperostosis significantly around the costovertebral joint, and that the restricted rib cage movement as a result of the ankylosis of thoracic spine will lead to further exacerbation of respiratory function in the case of cervical cord injury [9]. Moreover, it has been reported that elderly patients with DISH in the thoracic spine are susceptible to decreased respiratory function and have pneumonia due to the decline in flexibility of the thoracic cage [40,41]. Similarly, there was a significantly large number of cases in the DISH (+) group that had the hyperostosis of costovertebral joint in comparison to the DISH (−) group in this study. From this result, the data were analyzed to determine whether or not the ankylosis in the costovertebral joints or thoracic spine occurring as a result of DISH had any effect on the early mortality rate within 3 months of the initial examination, but there was no significant difference between the DISH (+) group and the DISH (−) group. This result could have been impacted by the fact that the target population in this study included many diseases except cervical cord injury, or the fact that the previous history of diabetes or cardiovascular disease was the same proportion in the two groups. However, the results do suggest the possibility that DISH itself will have no major impact on the prognosis for the pathology except cervical cord injury. However, a longer-term investigation is needed to confirm the significance of the mortality rate between these two groups in the future.

### 4.8. Limitations

Ideally, a prospective study would be conducted in order to adequately investigate the data, but this study was a retrospective study. The patient population included patients of all ages, but it was biased towards a disease group requiring emergency outpatient examination, rather than towards a generally healthy population. Furthermore, the investigation period was one year, resulting in a short-term patient evaluation. All of the co-authors were involved in the imaging evaluations, and no radiologists who do not know the clinical course of the patients after examination were included amongst the evaluators. The analysis of intra- and interobserver reliability for image evaluations was not performed. For emergency outpatients at our institute, CT imaging that includes the cervical region is not performed unless some sort of disease or trauma around the neck was suspected. For this reason, there was a limited number of cases that had CT imaging of the cervical spine in the one-year investigation, and the assessment of the cervical spine using CT imaging could not be included in this study. It will be necessary to conduct further research with more cases in the future, because the sample number of patients in this study was too small to achieve the good-quality evaluation of epidemiological study regarding DISH. Finally, since all of the individuals examined in this study were Japanese, the results fail to reflect other ethnicities. The results of this study must be interpreted in consideration of these limitations.

## 5. Conclusions

As a result of conducting an epidemiological survey on patients who underwent CT imaging from the chest to pelvis during examinations as emergency outpatients at our institution over a one-year period, the prevalence of DISH was 17.4% and these patients were predominantly elderly and male. When using the propensity score matching with the age and gender as confounding factors in order to compare the DISH (+) group and the DISH (−) group, there was no significant difference between these two groups in the previous medical history (diabetes mellitus, past cardiovascular disease), extent of aortic calcification or mortality rate within 3 months of the initial examination. The prevalence of hyperostosis around the costovertebral joint was only one variable which was found significantly in the DISH (+) group compared to the DISH (−) group. These results suggest that the majority of cases of DISH in the spinal column may involve the effects of age-related changes or biomechanical effects.

## Figures and Tables

**Figure 1 diagnostics-12-01088-f001:**
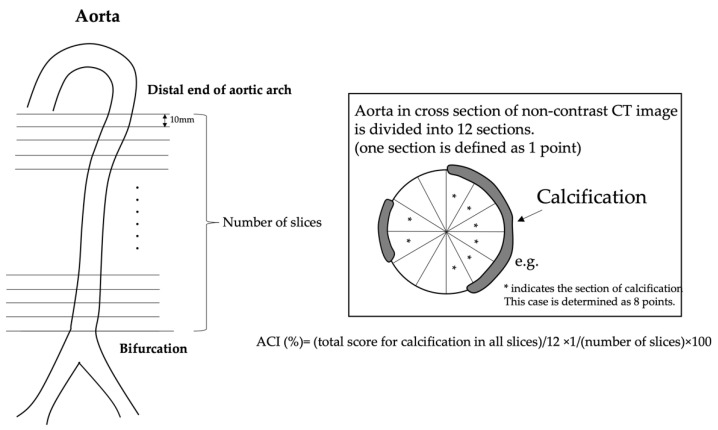
Evaluation of aortic calcification. The calcification of aorta was calculated from the distal end of aortic arch to the bifurcation of the aorta into the common iliac arteries. The cross section of the aorta on each slice was divided radially into 12 segments [12]. The aortic calcification index (ACI) was calculated as follows: ACI = (total score for calcification in all slices)/12 × 1/(number of slices) × 100 (%) [13,14].

**Figure 2 diagnostics-12-01088-f002:**
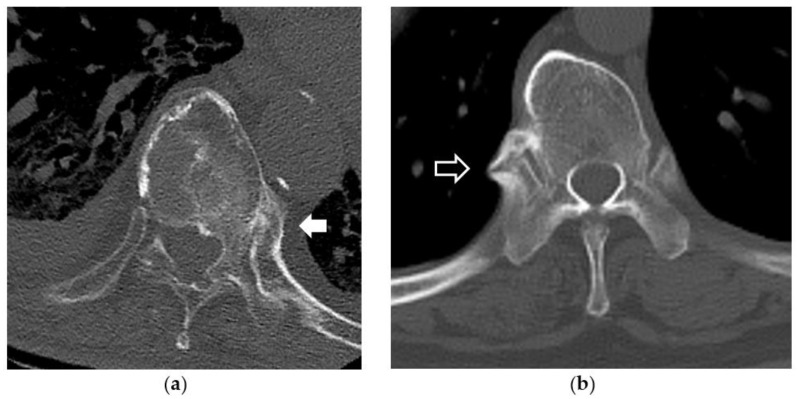
Evaluation of hyperostosis around the costovertebral joint. The cross-sectional CT image passing through the costovertebral joint from T1 vertebra to the T12 vertebra was used to evaluate the hyperostosis around the costovertebral joint. (**a**) There is ankylosis and excrescence in the left costovertebral joint (white arrow). (**b**) There is bone bridging in the right costovertebral joint (black arrow).

**Figure 3 diagnostics-12-01088-f003:**
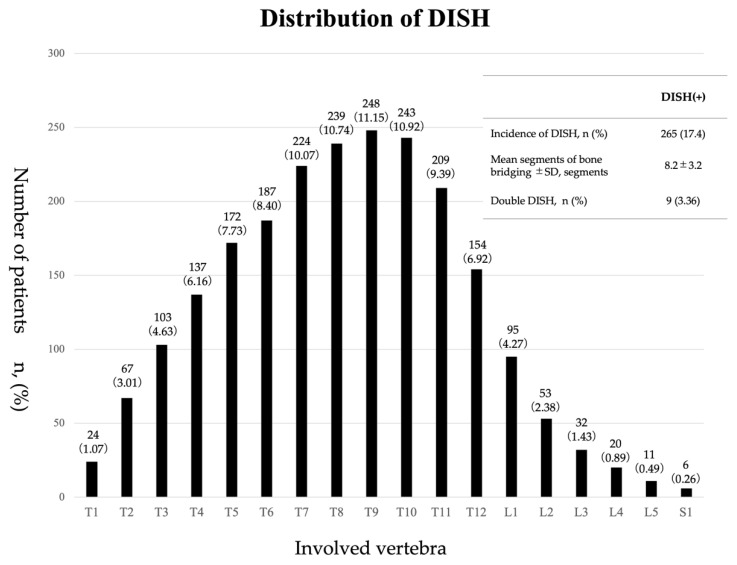
Distribution of DISH. The prevalence of DISH is 17.4%, with DISH observed in all of the segments from the T1 vertebra to the first sacral vertebra. The frequency of the occurrence in the thoracic spine was higher than in the lumbar spine, with a concentration in the thoracolumbar junction. The prevalence of DISH was particularly high (11.1%) in T9 vertebra. Bone bridging was observed in 8.2 ± 3.2 vertebrae on average, and there were 9 cases (3.3%) that showed double DISH.

**Table 1 diagnostics-12-01088-t001:** Patients’ demographic data.

	DISH (+) (*n* = 265, 17.4%)	DISH (−) (*n* = 1254, 82.5%)	*p*-Value	SMD
Mean age ± SD, yrs	78.8 ± 10.1	67.8 ± 19.5	<0.001 *	0.711
Gender, m/f, (% of m)	185/80 (69.8)	646/608 (51.6)	<0.001 *	0.387
BMI ± SD, kg/m^2^	22.1 ± 4.3	23.4 ± 3.7	0.378	NA

DISH, diffuse idiopathic skeletal hyperostosis; SMD, standardized mean difference; m, male; f, female; BMI, body mass index. *, statistically significant difference.

**Table 2 diagnostics-12-01088-t002:** Patients’ demographic data post sample matching using propensity score.

	DISH (+) (*n* = 188)	DISH (−) (*n* = 188)	*p*-Value	SMD
Mean age ± SD, yrs	78.8 ± 9.5	79.3 ± 9.5	0.564	0.060
Gender, m/f, (% m)	123/65 (65.4)	117/71 (62.2)	0.592	0.066
BMI ± SD, kg/m^2^	22.2 ± 4.4	20.5 ± 4.0	0.264	NA

DISH, diffuse idiopathic skeletal hyperostosis; SMD, standardized mean difference; m, male; f, female; BMI, body mass index.

**Table 3 diagnostics-12-01088-t003:** Medical history, ACI, costovertebral joint fusion and mortality.

	DISH (+) (*n* = 188)	DISH (−) (*n* = 188)	*p*-Value
Diabetes mellitus, *n* (%)	43 (22.9)	34 (18.1)	0.313
Medical history of cardiovascular disease, *n* (%)	95 (50.5)	86 (45.7)	0.352
Mean ACI ± SD, %	46.4 ± 30.0	47.2 ± 30.6	NA
Number of over 50% of ACI, *n* (%)	93 (49.5)	96 (51.1)	0.756
Number of hyperostosis around the costovertebral joint, *n* (%)	86 (45.7)	12 (6.3)	<0.001 *
Mortality within 3 weeks of the initial examination, *n* (%)	20 (10.6)	25 (13.3)	0.522

DISH, diffuse idiopathic skeletal hyperostosis; *n*, number; ACI, aortic calcification index. *, statistically significant difference.

**Table 4 diagnostics-12-01088-t004:** Medical history regarding cardiovascular disease.

	DISH (+), *n* (%)	DISH (−), *n* (%)	*p*-Value
Cerebral infarction	19 (15.5)	15 (12.1)	0.398
Cerebrovascular hemorrhage	5 (4.0)	10 (8.1)	0.188
Cardiac valvulopathy	3 (2.4)	2 (1.6)	0.644
Angina pectoris	9 (7.3)	10 (8.1)	0.825
Myocardial infarction	15 (12.2)	9 (7.3)	0.189
Atrial fibrillation	14 (11.4)	9 (7.3)	0.264
Chronic heart failure	8 (6.5)	12 (9.7)	0.360
Chronic renal failure	6 (4.9)	8 (6.5)	0.592
Hemodialysis	6 (4.9)	3 (2.4)	0.302
Hypertension	23 (18.8)	34 (27.6)	0.103
Pulmonary thromboembolism	1 (0.8)	2 (1.6)	0.566
Arteriosclerosis obliterans	1 (0.8)	1 (0.8)	0.995
Aortic aneurysm	9 (7.3)	3 (2.4)	0.073
Aortic dissection	3 (2.4)	5 (4.0)	0.479

DISH, diffuse idiopathic skeletal hyperostosis; *n*, number.

## Data Availability

The data used in this study are available upon reasonable request from the corresponding author.

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
