# Peer review of "The Prevalence and Characteristics of Diffuse Idiopathic Skeletal Hyperostosis (DISH): A Cross-Sectional Study of 1519 Japanese Individuals"

_diagnostics, 2022, doi:10.3390/diagnostics12051088_

Round 1

Reviewer 1 Report

This cross-sectional study revealed 17.4% of DISH in 1519 patients receiving thoracolumbar CT and indicated age, male and hyperostosis around the costovertebral joint were significantly higher in DISH (+) group. This paper is well-written and study design is reasonable. I have only one concern.

1. The inclusion criteria of this 1519 patients cohort are not clear. What kind of injury did they suffer from? or under what diagnosis did they receive thoracolumbar CT?

Author Response

For reviewer 1

Thank you very much for your reviewing for my manuscript.

Q1

  • The inclusion criteria of this 1519 patients cohort are not clear. What kind of injury did they suffer from? or under what diagnosis did they receive thoracolumbar CT?

I can’t explain the detail of diagnosis of patients in this study, because there are so many diagnosis in this series (internal medicine to trauma). So I changed the sentence like down below in line 64-67.

  • All of the patients that were examined with the torso computed tomography (CT) imaging as outpatients at the tertiary emergencymedical center of our institute from January 2020 to December 2020 were included regardless of their diagnosis in this cross-sectional study on DISH.

Reviewer 2 Report

Dear Author,

I've read the paper titled "The epidemiology of diffuse idiopathic skeletal hyperostosis (DISH): a cross-sectional study of 1519 Japanese individuals" with great interest.

The authors have reported on a cross-sectional epidemiological survey on DISH under thoracic spine that was conducted on patients who were examined as outpatients at a tertiary emergency medical center.  They have found 17.4% prevalence of DISH, and those patients were predominantly elderly and male.

I have some concerns relating to this well-written study:

Major Concerns:

The epidemiological terms used in the paper might make the readers confused.  The authors used both prevalence and incidence for the same issues (numbers), however, epidemiologically those terms are different and the authors should revisit the paper focusing on such issue.

As the study evaluated the radiological images, intra and interobserver reliability analyses should be performed for leveraging the scientific level of the study - if the authors are not willing to evalute such issue, this should be presented as a limitation.

As cited in the references, Hirasawa et al. evaluated the imaging studies (CT scans) of the patients at any department other than orthopedic surgery (http://dx.doi.org/10.1016/j.jos.2016.02.001), and presented a prevalence of 27.1% for DISH.  As the results were compared, the present study found 17.4% prevalence which was similar to the X-ray results of the study by Hirasawa et al.  Both patient populations were Japanese in origin, geographical conditions even may affect the study results.  This issue should be further evaluated and discussed.   

Such an epidemiologic study might include more parameters than prevalence, disease distribution, medical histories, and mortality... So, further statistical analyses should be performed by a statistician.

Thank you,

Respectfully yours,

Author Response

For reviewer 2

Thank you very much for your reviewing for my manuscript.

Q1

  • The epidemiological terms used in the paper might make the readers confused.  The authors used both prevalence and incidence for the same issues (numbers), however, epidemiologically those terms are different and the authors should revisit the paper focusing on such issue.

I changed the title like down below.

  • The prevalence and characteristics of diffuse idiopathic skeletal hyperostosis (DISH): a cross-sectional study of 1519 Japanese individuals

Q2

  • As the study evaluated the radiological images, intra and interobserver reliability analyses should be performed for leveraging the scientific level of the study - if the authors are not willing to evaluate such issue, this should be presented as a limitation.

I added the sentence in the section of limitation in line 370-372.

-  The analysis of intra and interobserver reliability for image evaluations was not performed.

Q3

  • As cited in the references, Hirasawa et al. evaluated the imaging studies (CT scans) of the patients at any department other than orthopedic surgery (http://dx.doi.org/10.1016/j.jos.2016.02.001), and presented a prevalence of 27.1% for DISH.  As the results were compared, the present study found 17.4% prevalence which was similar to the X-ray results of the study by Hirasawa et al.  Both patient populations were Japanese in origin, geographical conditions even may affect the study results.  This issue should be further evaluated and discussed. Such an epidemiologic study might include more parameters than prevalence, disease distribution, medical histories, and mortality... So, further statistical analyses should be performed by a statistician.

I added sentences in the section of Course of the epidemiological survey on DISH in line 271-276.

  • Hasegawa et al. reported the prevalence of 27.2% for DISH on CT imaging in Japanese individuals at any department other than orthopaedic surgery [28]. However, this study found the prevalence of 17.4 % on CT imaging. We consider that the difference of these results would be affected by the disease bias of an emergency outpatients and the lack of testing for intra and interobserver error (one time evaluation, and two residents and two senior spine surgeons for observation) in this study.

Round 2

Reviewer 2 Report

Dear Author,

I've read the revised manuscript "The prevalence and characteristics of diffuse idiopathic skeletal hyperostosis (DISH): a cross-sectional study of 1519 Japanese individuals" with great interest.

The authors have revised the manuscript regarding the reviewer reports.

I have one concern that could be further clarified:

I am not sure that this prevalence study could include "The evaluation of mortality rate within 3 months of the initial examination" with the given information in the discussion section.  Although this knowledge is important, I think there'd be a bias because most of the patients are more than 65 years of age, and had a reason to be evaluated in an emergency service.  So, such an issue should be better presented why the authors evaluate mortality rate within 3 months after the initial examination.

p12 l362: typo error: beteween

Thank you,

Respectfully yours,

Author Response

For reviewer 2

Thank you very much for your comment in the second round of reviewing for my manuscript.

Q1.

  • I am not sure that this prevalence study could include "The evaluation of mortality rate within 3 months of the initial examination" with the given information in the discussion section.  Although this knowledge is important, I think there'd be a bias because most of the patients are more than 65 years of age, and had a reason to be evaluated in an emergency service.  So, such an issue should be better presented why the authors evaluate mortality rate within 3 months after the initial examination.

I added the sentences in the section of Discussion (4.7. Mortality rate within 3 months of the initial examination) in line 345-348.

  • In this study, we set the period of the mortality rate to 3 months after an admission, because the clinical courses of patients who hospitalized with a severe general condition at the tertiary emergency medical center is often aggravated within 2 months, moreover the mean age of patients that included in this study were over 65 years.